# Impact of maternal education on response to lifestyle interventions to reduce gestational weight gain: individual participant data meta-analysis

Eileen C O'Brien,[1] Ricardo Segurado,[2] Aisling A Geraghty,[1] Goiuri Alberdi,[1] Ewelina Rogozinska,[3,4] Arne Astrup,[5] Rubenomar Barakat Carballo,[6] Annick Bogaerts,[7,8,9] Jose Guilherme Cecatti,[10] Arri Coomarasamy,[11] Christianne J M de Groot,[12] Roland Devlieger,[7,13] Jodie M Dodd,[14] Nermeen El Beltagy,[15] Fabio Facchinetti,[16] Nina Geiker,[17] Kym Guelfi,[18] Lene Haakstad,[19] Cheryce Harrison,[20] Hans Hauner,[21] Dorte M Jensen,[22] Khalid Khan,[3,4] Tarja Inkeri Kinnunen,[23] Riitta Luoto,[23] Ben Willem Mol,[24] Siv Mørkved,[25,26] Narges Motahari-Tabari,[27] Julie A Owens,[14] Maria Perales,[6] Elisabetta Petrella,[16] Suzanne Phelan,[28] Lucilla Poston,[29] Kathrin Rauh,[30] Girish Rayanagoudar,[3] Kristina M Renault,[31,32] Anneloes E Ruifrok,[33] Linda Sagedal,[34] Kjell Å Salvesen,[35,36] Tania T Scudeller,[37] Gary Shen,[38] Alexis Shub,[39] Signe N Stafne,[25,26] Fernanda G Surita,[10] Shakila Thangaratinam,[3,4] Serena Tonstad,[40] Mireille N M van Poppel,[41] Christina Vinter,[42] Ingvild Vistad,[34] SeonAe Yeo,[43] Fionnuala M McAuliffe,[1] i-WIP (International Weight Management in Pregnancy) Collaborative Group

For numbered affiliations see end of article.

**Correspondence to**
Professor Fionnuala M McAuliffe;
fionnuala.mcauliffe@ucd.ie

## ABSTRACT

**Objectives** To identify if maternal educational attainment is a prognostic factor for gestational weight gain (GWG), and to determine the differential effects of lifestyle interventions (diet based, physical activity based or mixed approach) on GWG, stratified by educational attainment.

**Design** Individual participant data meta-analysis using the previously established International Weight Management in Pregnancy (i-WIP) Collaborative Group database (https://iwipgroup.wixsite.com/collaboration). Preferred Reporting Items for Systematic reviews and Meta-Analysis of Individual Participant Data Statement guidelines were followed.

**Data sources** Major electronic databases, from inception to February 2017.

**Eligibility criteria** Randomised controlled trials on diet and physical activity-based interventions in pregnancy. Maternal educational attainment was required for inclusion and was categorised as higher education (≥tertiary) or lower education (≤secondary).

**Risk of bias** Cochrane risk of bias tool was used.

**Data synthesis** Principle measures of effect were OR and regression coefficient.

**Results** Of the 36 randomised controlled trials in the i-WIP database, 21 trials and 5183 pregnant women were included. Women with lower educational attainment had an increased risk of excessive (OR 1.182; 95% CI 1.008 to 1.385, p =0.039) and inadequate weight gain (OR 1.284; 95% CI 1.045 to 1.577, p =0.017). Among women with lower education, diet basedinterventions reduced risk of excessive weight gain (OR 0.515; 95% CI 0.339 to 0.785,

p = 0.002) and inadequate weight gain (OR 0.504; 95% CI 0.288 to 0.884, p=0.017), and reduced kg/week gain (B −0.055; 95% CI −0.098 to −0.012, p=0.012). Mixed interventions reduced risk of excessive weight gain for women with lower education (OR 0.735; 95% CI 0.561 to 0.963, p=0.026). Among women with high education, diet based interventions reduced risk of excessive weight gain

## Strengths and limitations of this study

▶ A major strength is the use of individual participant data to carry out meta-analysis from multiple nutrition and lifestyle interventions.

▶ The pooling of individual participant data allows for the estimation of the differential effects of education and lifestyle interventions on weight gain across the included cohorts with greater power than individual randomised controlled trials.

▶ We analysed and reported the findings using gestational weight gain per week, which enables the comparison of studies that used different gestational ages when total weight gain was measured.

▶ A limitation is that most of the cohorts included are from wealthy countries; therefore, the results may not be representative of populations living in low-income and middle-income countries.

▶ Although the results were statistically significant, the effect sizes are small; thus, the relevance of these findings in the clinical setting must be considered.

(OR 0.609; 95% CI 0.437 to 0.849, p=0.003), and mixed interventions reduced kg/week gain (B −0.053; 95% CI −0.069 to −0.037,p<0.001). Physical activity based interventions did not impact GWG when stratified by education.

**Conclusions** Pregnant women with lower education are at an increased risk of excessive and inadequate GWG. Diet based interventions seem the most appropriate choice for these women, and additional support through mixed interventions may also be beneficial.

## INTRODUCTION

According to the 2030 Sustainable Development Goals, the global health agenda is calling for improvements in maternal health and nutrition, equal opportunities for education and reduced social, economic and political inequalities.[1] The current situation is that maternal social, economic and educational backgrounds are key determinants of health and lifestyle behaviours among pregnant women across the world. Pregnancy and birth outcomes are negatively influenced by socioeconomic and deprivation factors.[2–6] Prevalence of smoking, alcohol consumption and prepregnancy body mass index (BMI) are greater among women of lower socioeconomic status, while diet quality and physical activity participation are less within this group.[7–9]

Gestational weight gain (GWG) is also influenced by socioeconomic status, with a recent systematic review finding that those of lower educational attainment are less likely to gain weight within the guidelines' recommended ranges for GWG, as described by the Institute of Medicine (IOM).[10] Excessive GWG increases the risk of adverse maternal and neonatal outcomes, including gestational diabetes, caesarean section, hypertension, large for gestational age and stillbirth, while inadequate GWG is associated with increased risk of small-for-gestational-age infant.[11 12] Moreover, greater GWG is associated with weight retention postpartum[13] and elevated BMI up to 15 years post pregnancy,[14] a potential factor exacerbating current global trends of overweight and obesity.[15]

Since GWG outside of suggested guidelines is more prevalent among disadvantaged groups, it would seem wise that support and interventions that enable equitable opportunities to optimise health are prioritised. However, knowledge of the impact of interventions on GWG among women of various socioeconomic position is limited. Pregnancy has been described as a stimulus for positive behaviour change[16] and is a time when women are in regular contact with healthcare professionals.[17] Thus, a lifestyle intervention in pregnancy should, in theory, have a greater impact during this time than any other time in a woman's life. While there are many examples of lifestyle and behaviour interventions in the literature that aim to manage weight gain during pregnancy,[18] few interventions stratify analyses by socioeconomic status.

The International Weight Management in Pregnancy (i-WIP) Individual Patient Data Collaborative Network was established in 2013 to determine the differential effects of weight management interventions in pregnancy on maternal weight gain and pregnancy outcomes.[19] The collaboration between researchers from 16 countries has resulted in a repository of data from 36 randomised trials with over 60 variables and individual data from more than 12 500 pregnant women. The i-WIP found that diet and physical activity-based interventions reduce GWG and lower the odds of caesarean section.[20] However, subanalysis was not conducted among women of various educational attainments, thus compounding our lack of understanding for the type of pregnancy intervention that has the greatest effect among those of low educational attainment.

This research aims to use individual participant data (IPD) meta-analysis using data from the previously established i-WIP Network to: (1) identify if educational attainment is a prognostic factor for GWG; (2) determine the differential effects of weight management interventions in pregnancy on maternal weight gain, stratified by educational attainment.

## METHODS

The checklist outlined in the Preferred Reporting Items for Systematic Reviews and Meta-Analysis of Individual Participant Data (PRISMA-IPD) statement was followed.[21] Full information about the search strategy is available in the i-WIP protocol paper and the i-WIP primary publication.[19 20] In brief, major electronic databases from October 2013 to March 2015 were searched, with updated literature searches completed by the i-WIP team in February 2017 and more recently in June 2018. Given the logistics and resources involved in collating IPD, trials identified in the most recent literature search have not yet been added to the i-WIP dataset. Eligible studies were randomised controlled trials (RCT) that evaluated any dietary or lifestyle interventions with potential to influence maternal and fetal outcomes related to maternal weight.[19 22] The women in the study were required to be ≥18 years of age with a singleton pregnancy and they must have been enrolled to one of the following interventions: diet based, physical activity based, mixed approach (diet, physical activity, GWG monitoring and/or behaviour change components) or standard care.[22] In total, 11 920 studies were identified, of which 36 studies were included (12 526 participants).[19 20 23] The flow chart for the identification and selection of studies has been published[20] (online supplementary appendix 1). A full search strategy from the original i-WIP IPD for one database is presented in online supplementary appendix 2. All individual trials included in this IPD meta-analysis had ethical approval from relevant ethical committees.

### IPD integrity and risk-of-bias assessment

The i-WIP Network performed a range of checks on the variables used during the analysis.

The randomisation ratio, baseline characteristics and method of analysis provided by the original authors to the i-WIP Network were compared with the published

information. Any discrepancies were queried and rectified as necessary with input from the original authors. As per the protocol, the i-WIP Network assessed the quality of each trial to evaluate the integrity of the randomisation and follow-up procedure. The Cochrane risk-of-bias tool was used and studies were classified as high risk of bias if they scored highly in at least one of following domains: randomisation, allocation concealment, blinding of outcome assessment or incomplete outcome data. To be classified as low risk of bias, all items must have scored as low risk. In terms of publication bias, a funnel plot was created to assess the small study effects on trials in the IPD meta-analysis of educational attainment and GWG per week (online supplementary figure S1).

### Data items and specification of outcomes

The i-WIP protocol prespecified that socioeconomic status would be evaluated as a potential prognostic variable for GWG, as secondary analysis to the i-WIP project.[19] The exposure variable was maternal educational attainment, as this was the most commonly used measure of socioeconomic status among studies in the i-WIP network. It has also been reported as a stable measure of individual socioeconomic status.[24] Educational attainment was classified as less than or equal to secondary education (including no education, less than primary, completed primary, some secondary or completed secondary education), or tertiary education (including vocational training, some university, bachelor degree, postgraduate degree or higher).

Intervention type was classified as diet based, physical activity based or mixed approach (diet, physical activity, GWG monitoring and/or behaviour change components)[22] and participants were either exposed to the intervention or to standard care.

The primary outcome was maternal weight gain in pregnancy. The outcome was examined using two methods: according to the IOM 2009 guidelines[25]; and GWG per week as a continuous variable. To calculate GWG, data were required for the following variables: baseline gestational age, final follow-up gestational age, prepregnancy or early pregnancy weight, final follow-up weight and height. The method described in online supplementary appendix 3 was used to calculate GWG per week.

### Specification of effect measures

The principle measures of effect were OR (for analysis according to the IOM guidelines) and regression coefficient (for analysis of GWG as a continuous variable).

To assess the magnitude of study variability in the outcomes, we calculated an intraclass correlation coefficient (ICC) from the fitted models, interpretable as an index of heterogeneity such as the I$^2$. The intraclass correlation coefficient for the study ($ICC_{study}$) relates to the proportion of variability due to cohort-to-cohort variation, and the intraclass correlation coefficient for the intervention ($ICC_{intervention}$) relates to the proportion of variability in the intervention treatment effect. In the

logistic models for categorical GWG, the residual variance was replaced by $\varpi^2/3$.

### Synthesis methods

For comparisons of characteristics across education groups, we used independent sample t-tests for continuous data and $\chi^2$ tests for categorical data. For the categorical outcomes of inadequate, adequate and excessive GWG, we used generalised linear mixed models: a multinomial logistic regression with a generalised logit link function. For the continuous outcomes of GWG (kg/week), we used a linear mixed model with an identity link function, assuming the outcome was approximately normally distributed. The models analysing the effect of education included random effects for each cohort. Random effects were specified for both the intercept and the slope of the intervention effect, allowing them to be correlated. The between-study variance in each effect was calculated by the ICC. Separate models were created for educational attainment strata to analyse the effect of each intervention type, as fixed effects. Linear mixed models were adjusted for factors known to influence GWG (BMI, maternal age, parity, ethnicity), in addition to the main effects and interaction of education and the interventions. In model 1, we adjusted for prepregnancy maternal BMI (continuous GWG model only). In model 2, we adjusted for prepregnancy maternal BMI (continuous GWG model only), maternal age, parity and ethnicity. As the IOM category definitions are linked to prepregnancy BMI category, this outcome intrinsically 'adjusts' for maternal BMI; we can consider model 1 as 'unadjusted', and model 2 as an 'adjusted' model. All statistical analyses were performed using IBM SPSS Statistics for Windows, V.24.0 (IBM).

### Exploration of variation in effects

While controlling for between-study variation in outcomes and intervention effects as random effects, we explored the influence of educational attainment on the intervention effects as prespecified fixed main effects and an interaction term. Tertiary education and the control (non-intervention) condition were set as the reference levels, and results for IOM categories used the adequate weight gain category as the reference outcome group.

### Additional analyses

We proceeded with our model building in a stepwise fashion, aiming at a prespecified target of fixed effects as above, and both random intercept and intervention effects. A stepwise approach is often pragmatic in developing a mixed-effect model as small study size and uncertain variance/covariance structure can lead to models failing to converge. We therefore combined the addition of random and fixed effects to a model in the following stages: (1) empty model including no fixed predictors, only a random intercept; (2) education and intervention main effects and interaction added to the model, and controlling for prepregnancy maternal BMI (model

**Table 1** Participant exclusion criteria for individual participant data

| Reason for exclusion | Excluded N | Remaining N |
|---|---|---|
| Full dataset | | 12 240 |
| Data missing: baseline gestational age | 2611 | 9629 |
| Data missing: final follow-up gestational age | 2309 | 7320 |
| Data missing: maternal educational attainment | 1878 | 5442 |
| Multiple pregnancy | 17 | 5425 |
| Less than 18 years of age | 75 | 5350 |
| Data missing: prepregnancy and early pregnancy weight | 1 | 5349 |
| Data missing: final follow-up weight | 137 | 5212 |
| Data missing: height (for prepregnancy BMI) | 29 | 5183 |

BMI, body mass index.

1); (3) random effect of intervention was explored to account for potential population or intrinsic variability in the effect of the interventions on GWG; (4) additional potential confounders were added as individual-level fixed effects to either model 2 or 3: maternal age, parity and ethnicity (model 2). Our strategy was to remove fixed interaction effects or random effects if not statistically significant, or if model fitting became problematic. Furthermore, within each study, independent sample t-tests were carried out to explore the differences in weight gain per week by education level.

### Patient involvement
No patients were involved in the design of this study. However, we plan to share and disseminate the findings with one of the parent participant involvement groups (the ROLO (Randomised contol trial of low glycaemic index diet to prevent the recurrence of macrosomia) Families Advisory Committee).

### RESULTS
### Protocol, IPD obtained and study characteristics
The PRISMA-IPD checklist is included as a online supplementary file. The original i-WIP IPD dataset contained 12 240 participants from 36 studies. Participants were excluded if they were missing data relating to the main outcomes or met the exclusion criteria (table 1).

In total, 7057 women were excluded from the analysis, while 5183 remained. Participants that met the inclusion criteria were enrolled to 21 RCTs.[26–46] The study characteristics, methods used to measure GWG, intervention details and outcome measures of the eligible trials are summarised in online supplementary table S1.

### IPD integrity and risk-of-bias assessment
No issues were identified in the integrity of the IPD. The global classification of risk of bias classified 12 studies as high risk and 9 as low/medium risk (online supplementary table S2). In terms of publication bias, the funnel plot (online supplementary figure S1) demonstrated an even distribution of effect estimates and SE, indicating low bias in publications.

### Results of individual studies
For the majority of studies, we did not observe a significant difference in GWG/week (kg) between participants in the intervention and control, split by educational attainment (online supplementary table S3). Within four of the studies,[28 34 41 45] women with tertiary education enrolled to the intervention had significantly less weight gain/week. One study found that women with secondary education or less enrolled to the intervention had significantly less weight gain/week.[46] The forest plot (online supplementary figure S2) demonstrates the results of individual studies for the IPD meta-analysis of low educational attainment and GWG per week, controlling for BMI and intervention. A comparison of maternal characteristics among women included and excluded in the IPD meta-analysis (online supplementary table S4) demonstrated some differences across the populations.

### Results of syntheses
Of the 5183 participants in this analysis, 1910 (36.9%) had, at most, completed secondary education, while 3273 (63.1%) had achieved at least some tertiary education . Within the secondary education group, 931 (48.8%) were enrolled to the control arm, and 653 (34.2%), 186 (9.7%) and 140 (7.3%) were enrolled to the intervention arm of mixed, diet and physical activity trials, respectively. Within the tertiary education group, 1531 (46.8%) were enrolled to the control arm of their respective RCT, while 1023 (31.2%), 323 (9.9%) and 396 (12.1%) were enrolled to the intervention arm of mixed, diet and physical activity trials, respectively.

Compared with women with tertiary education, women with lower education levels had significantly higher early pregnancy weight and prepregnancy BMI, and were significantly shorter in height (table 2). Fewer women with tertiary education had a BMI>30 kg/m$^2$, compared with women in the secondary education group (26.8% vs 46.3%, p<0.001). Women within the tertiary education group were older, had lower parity and were more likely to be of Caucasian ethnicity compared with those with lower education.

Mean±SD GWG per week and percentage of participants within each of the IOM categories, stratified by educational attainment and BMI category are shown in table 3. Figure 1 displays GWG (kg/week), according to prepregnancy BMI, intervention and educational attainment.

The unadjusted linear mixed models found that women with lower educational attainment gained significantly

**Table 2** Participant characteristics in the total group, and split by education category

| | Total | | Lower education | | Higher education | | |
|---|---|---|---|---|---|---|---|
| | **N** | | **N** | | **N** | | **P value*** |
| Early pregnancy weight (kg)† | 5183 | 78.60 (18.56) | 1910 | 82.51 (20.26) | 3273 | 76.31 (17.08) | <0.001 |
| Prepregnancy weight (kg)† | 5183 | 76.06 (18.74) | 1712 | 79.61 (20.37) | 3176 | 73.17 (16.96) | <0.001 |
| Height (cm)† | 5183 | 166.07 (7.03) | 1910 | 164.76 (7.17) | 3273 | 166.84 (6.84) | <0.001 |
| Prepregnancy BMI (kg/m$^2$)† | 5183 | 27.57 (6.55) | 1910 | 29.45 (7.08) | 3273 | 26.47 (5.95) | <0.001 |
| Underweight‡ | 5183 | 77 (1.5) | 1910 | 32 (1.6) | 3273 | 46 (1.4) | <0.001 |
| Normal weight‡ | | 2274 (43.9) | | 603 (31.5) | | 1672 (51.1) | |
| Overweight‡ | | 1073 (20.7) | | 393 (20.5) | | 681 (20.8) | |
| Obese‡ | | 1763 (34.0) | | 886 (46.3) | | 878 (26.8) | |
| Gestational age (weeks) | | | | | | | |
| Baseline§ | 5183 | 14.5 (4.0–32.0) | 1910 | 13.9 (4.0–32.0) | 3273 | 14.8 (4.0–32.0) | <0.001 |
| Final follow-up§ | 5183 | 36.6 (24.0–44.0) | 1910 | 36.6 (27.0–44.0) | 3273 | 36.5 (24–43.0) | 0.196 |
| Delivery§ | 5172 | 39.7 (26.0–44.0) | 1907 | 39.50 (27.0–44.0) | 3265 | 39.75 (26.0–43.0) | <0.001 |
| Infant birth weight (g)† | 5164 | 3531.73 (547.90) | 1908 | 3476.55 (559.43) | 3256 | 3564.06 (538.49) | <0.001 |
| Maternal age (years)† | 5183 | 29.90 (4.97) | 1910 | 28.50 (5.51) | 3273 | 30.72 (4.43) | <0.001 |
| Parity† | 5135 | 0.73 (1.05) | 1884 | 0.93 (1.20) | 3251 | 0.62 (0.93) | <0.001 |
| Ethnicity | | | | | | | |
| Caucasian‡ | 3719 | 3291 (63.5) | 1203 | 954 (49.9) | 2516 | 2338 (71.4) | <0.001 |
| Asian‡ | | 84 (1.6) | | 25 (1.3) | | 60 (1.8) | |
| Black‡ | | 123 (2.4) | | 71 (3.7) | | 53 (1.6) | |
| Central/South American‡ | | 88 (1.7) | | 76 (3.9) | | 13 (0.4) | |
| Middle East (including Iran and Turkey)‡ | | 63 (1.2) | | 45 (2.3) | | 19 (0.5) | |
| Other‡ | | 76 (1.4) | | 38 (1.9) | | 39 (1.2) | |
| Missing‡ | | 1465 (28.2) | | 708 (37.0) | | 758 (23.1) | |

Lower education: secondary education or less. Higher education: at least some tertiary education.
Underweight: <18.5 kg/m$^2$; normal, 18.5–24.9 kg/m$^2$; obese, ≥30 kg/m$^2$; overweight, 25.0–29.9 kg/m$^2$.
*P value: analysis of variance and $\chi^2$.
†Data are presented as mean (SD).
‡Data are presented as n (%).
§Data are presented as mean (range of minimum and maximum).
BMI, body mass index.

more weight per week (kg) compared with those with tertiary education (B=0.014); however, this association was not significant when the model was adjusted for maternal age, parity and ethnicity (table 4a). The ICC$_{study}$ was 0.1% and ICC$_{intervention}$ was 0.0%. Women of lower educational attainment had an increased risk of both excessive (OR 1.182) and inadequate (OR 1.284) GWG as per the IOM guidelines, controlling for intervention, maternal age, parity, ethnicity, with random effects (table 4b). The ICC calculations for model 2 of table 4b were low; excessive weight gain: ICC$_{study}$ 8.1% and ICC$_{intervention}$ 0.4%; inadequate weight gain: ICC$_{study}$ 5.0% and ICC$_{intervention}$ 0.7%.

The intervention-by-education interaction effects were not statistically significant, and for ease of interpretation,

these were removed; thus, we present results for models stratified by education.

Among those of lower educational attainment, a significant reduction in weight gain per week (kg) was observed for both diet based interventions (B=−0.055) and mixed interventions (B=−0.044), compared with those in the control arms, table 5a. No significant effect was observed for physical activity asedalone interventions. Women with tertiary education enrolled to mixed interventions gained significantly less GWG per week (kg) (B=−0.053), than those in the control arms. Neither diet-alone nor physical activity based interventions had significant effects on weight gain per week (kg) for women with tertiary education. The adjusted models (model 2) included BMI,

**Table 3** Gestational weight gain (GWG) per week (kg/week), and per Institute of Medicine (IOM) guidelines, body mass index (BMI) category and education

| | BMI category | N | GWG/week (kg) Total GWG/week Mean (SD) | Trimesters 2 and 3 GWG/week (kg), and per IOM guidelines | | | |
|---|---|---|---|---|---|---|---|
| | | | | T2 and 3 GWG/week Mean (SD) | Inadequate, n (%) | Adequate, n (%) | Excess, n (%) |
| All participants | Total | 5183 | 0.45 (0.22) | 0.46 (0.22) | 836 (16.1%) | 1224 (23.6%) | 3123 (60.3%) |
| | Underweight | 76 | 0.50 (0.16) | 0.51 (0.16) | 23 (30.3%) | 34 (44.7%) | 19 (25.0%) |
| | Normal weight | 2273 | 0.51 (0.28) | 0.51 (0.18) | 347 (15.3%) | 778 (34.2%) | 1148 (50.5%) |
| | Overweight | 1072 | 0.48 (0.22) | 0.49 (0.22) | 113 (10.5%) | 128 (11.9%) | 831 (77.5%) |
| | Obese | 1762 | 0.35 (0.23) | 0.36 (0.24) | 353 (20.0%) | 284 (16.1%) | 1125 (63.8%) |
| Lower education | Total | 1910 | 0.42 (0.24) | 0.43 (0.25) | 363 (19.0%) | 401 (21.0%) | 1146 (60.0%) |
| | Underweight | 31 | 0.55 (0.20) | 0.56 (0.20) | 7 (22.6%) | 11 (35.5%) | 13 (41.9%) |
| | Normal weight | 602 | 0.51 (0.20) | 0.52 (0.21) | 108 (17.9%) | 184 (30.6%) | 310 (51.5%) |
| | Overweight | 392 | 0.46 (0.24) | 0.47 (0.24) | 55 (14.0%) | 59 (15.1%) | 278 (70.9%) |
| | Obese | 885 | 0.34 (0.24) | 0.35 (0.25) | 193 (21.8%) | 147 (16.6%) | 545 (61.6%) |
| Higher education | Total | 3273 | 0.46 (0.20) | 0.47 (0.21) | 473 (14.5%) | 823 (25.1%) | 1977 (60.4%) |
| | Underweight | 45 | 0.46 (0.10) | 0.47 (0.11) | 16 (35.6%) | 23 (51.1%) | 6 (13.3%) |
| | Normal weight | 1671 | 0.50 (0.17) | 0.51 (0.17) | 239 (14.3%) | 594 (35.5%) | 838 (50.1%) |
| | Overweight | 680 | 0.49 (0.20) | 0.50 (0.20) | 58 (8.5%) | 69 (10.1%) | 553 (81.3%) |
| | Obese | 877 | 0.35 (0.23) | 0.37 (0.24) | 160 (18.2%) | 137 (15.6%) | 580 (66.1%) |

Lower education: secondary education or less. Higher education: at least some tertiary education.
Underweight: <18.5 kg/m$^2$; normal, 18.5–24.9 kg/m$^2$; obese, ≥30 kg/m$^2$; overweight, 25.0–29.9 kg/m$^2$.

maternal age, parity and ethnicity as confounders, with fixed effects.

Women with lower education were significantly less likely to exceed the IOM guidelines for weight gain per week if they were enrolled to diet based interventions (OR 0.515) or mixed interventions (OR 0.735), compared with those in the control arms, table 5b. These women also

had a reduced risk of inadequate weight gain per week if they were enrolled to a diet based intervention (OR 0.504). Women with tertiary education enrolled to dietary interventions were less likely to exceed IOM guidelines (OR 0.609); however, those enrolled to mixed interventions were at an increased risk of inadequate weight gain per week (OR 1.411), compared with those in the control

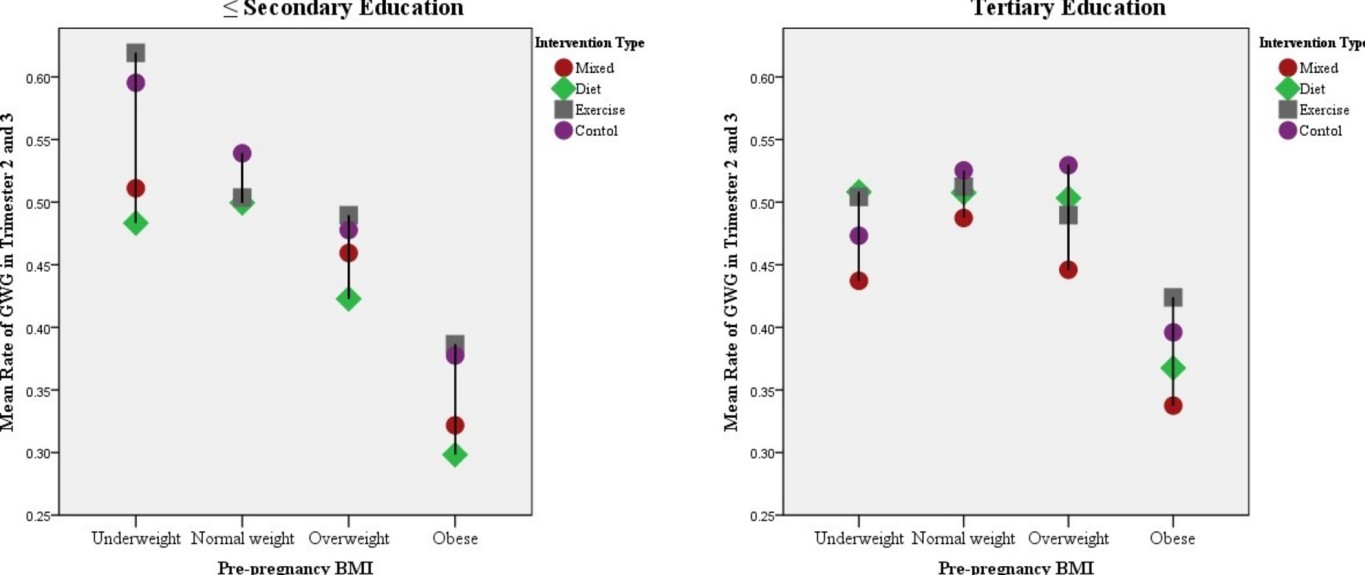

**Figure 1** Gestational weight gain (GWG) (kg/week), stratified by prepregnancy body mass index (BMI), intervention and educational attainment.

**Table 4** Generalised linear mixed models to show relationship between maternal educational attainment and (a) gestational weight gain per week (kg), and (b) gestational weight gain per week as per the Institute of Medicine (IOM) guidelines

| | | (a) Gestational weight gain per week (kg/week) | | (b) Gestational weight gain per week as per the IOM guidelines | | | |
| | | | | Excessive weight gain | | Inadequate weight gain | |
| | Education | B (95% CI) | P value* | OR (95% CI) | P value* | OR | P value* |
|---|---|---|---|---|---|---|---|
| Model 1 (n=5183) † | Lower | 0.014 (0.001 to 0.027) | 0.032 | 1.224 (1.052 to 1.425) | 0.009 | 1.259 (1.033 to 1.535) | 0.022 |
| | Higher | Ref | | Ref | | Ref | |
| Model 2 (n=5135) ‡ | Lower | 0.005 (−0.008 to 0.018) | 0.465 | 1.182 (1.008 to 1.385) | 0.039 | 1.284 (1.045 to 1.577) | 0.017 |
| | Higher | Ref | | Ref | | Ref | |

Lower education: secondary education or less. Higher education: at least some tertiary education.

*P value: Wald test.

†Adjusted for intervention with random intercept and intervention slope. Weight gain per week (kg/week) additionally adjusted for body mass index (BMI) (kg/m$^2$).

‡Adjusted for intervention, maternal age, parity, ethnicity, with random intercept and intervention slope. Weight gain per week (kg/week) additionally adjusted for BMI (kg/m$^2$).

arms. No effect was observed for physical activity based interventions. The adjusted models (model 2) included maternal age, parity and ethnicity as confounders, with fixed effects.

## DISCUSSION
### Summary of evidence
This IPD meta-analysis of the i-WIP consortium of lifestyle interventions in pregnancy found that low maternal education is associated with increased risk of both excessive and inadequate GWG per week, although crude differences were not observed in the percentage of women who gained inadequately, adequately and excessively between women of high and low educational attainment. Dietary-interventions reduced the risk of excessive GWG for all women, and reduced the risk of inadequate GWG for those with lower educational attainment. Mixed interventions were protective against excessive GWG for women with low education, but increased the risk of inadequate GWG among women with higher educational attainment. Physical activity based interventions were not associated with GWG, regardless of maternal education status.

### Compared with the literature
Maternal educational attainment less than or equal to secondary level was associated with an 18% increased risk of excessive GWG and a 28% increased risk of inadequate GWG per week, as per the IOM 2009 guidelines,[25] compared to those with at least some tertiary education. A systematic review of observational studies also found that women with low maternal educational attainment were less likely to gain appropriate weight, as per the 2009 IOM recommendations.[10] However, the review only provided a narrative of the evidence to date, not a meta-analysis, thus was restricted from drawing firm conclusions. Nonetheless, both the systematic review and our findings highlight the need for additional support for pregnant women of low educational attainment. Although the increased risk of inadequate and excessive GWG among women with

lower education was statistically significant, this is not synonymous with clinical significance. Between women of low and high educational attainment, the actual percentage differences in those who gained inadequately (19.0% vs 14.5%), adequately (21.0% vs 25.1%) and excessively (60.0% vs 60.4%) were relatively small.

The impact of interventions in pregnancy on weight gain has been previously examined[18 20 22 47]; however, analyses were not stratified by socioeconomic status within these publications. Thus, our study provides new insights in this field. A 2015 Cochrane review compared the effect of interventions in pregnancy (diet or exercise, or both) on GWG. Unlike our findings, in which only diet and mixed interventions reduced excessive weight gain, all categories of interventions significantly reduced excessive weight gain compared with control; diet (relative risk (RR) 0.77), diet and exercise (RR 0.86), and diet counselling/other (RR 0.46).[47] However, the review also highlighted an increased risk of low weight gain with all types of interventions; diet (RR 1.24), diet and exercise (RR 1.25), and diet counselling/other (RR 1.21).[47] A systematic review by Thangaratinam *et al* found that dietary interventions reduced total GWG by 3.84 kg, while mixed and physical activity interventions reduced total GWG to lesser extents (−1.06 kg and −0.72 kg, respectively).[22] Both mixed and physical activity interventions reduced risk of excessive weight gain, but insufficient data were available to assess the impact of dietary interventions on excessive weight gain.[22] The i-WIP primary publication in 2017 found that diet and physical activity-based interventions during pregnancy reduced GWG (−0.72 kg and −0.73 kg, respectively), but did not find differences in GWG across the subgroups examined (educational attainment was not assessed).[20]

In summary, based on our findings and previous studies, it seems that among women of high and low educational status, dietary interventions are most effective in promoting appropriate GWG. Mixed interventions could also be considered for those with lower education. Our

**Table 5** Generalised linear mixed models to show relationship between maternal educational attainment, type of intervention and (a) gestational weight gain per week (kg), and (b) gestational weight gain per week as per the Institute of Medicine (IOM) guidelines

| | | | (a) Gestational weight gain per week (kg/week) | | (b) Gestational weight gain per week as per the IOM guidelines | | | |
| | | | | | Excessive weight gain | | Inadequate weight gain | |
| | Education | Intervention type | B (95% CI) | P value* | OR (95% CI) | P value* | OR (95% CI) | P value* |
|---|---|---|---|---|---|---|---|---|
| Model 1† | Lower (n=1910) | Mixed approach | −0.044 (−0.069 to −0.019) | <0.001 | 0.745 (0.572 to 0.971) | 0.029 | 1.039 (0.745 to 1.449) | 0.821 |
| | | Diet based | −0.053 (−0.097 to −0.009) | 0.018 | 0.524 (0.351 to 0.782) | 0.002 | 0.496 (0.285 to 0.864) | 0.013 |
| | | Physical activity based | −0.020 (−0.069 to 0.029) | 0.422 | 0.704 (0.446 to 1.110) | 0.131 | 0.634 (0.333 to 1.208) | 0.166 |
| | | Control | Ref | | Ref | | Ref | |
| | Higher (n=3273) | Mixed approach | −0.056 (−0.071 to to −0.040) | <0.001 | 0.930 (0.753 to 1.150) | 0.504 | 1.407 (1.057 to 1.874) | 0.019 |
| | | Diet based | −0.015 (−0.038 to 0.009) | 0.213 | 0.619 (0.446 to 0.858) | 0.004 | 0.943 (0.565 to 1.574) | 0.822 |
| | | Physical activity based | −0.014 (−0.036 to 0.008) | 0.206 | 1.027 (0.753 to 1.402) | 0.865 | 1.209 (0.770 to 1.899) | 0.410 |
| | | Control | Ref | | Ref | | Ref | |
| Model 2‡ | Lower (n=1884) | Mixed approach | −0.044 (−0.069 to −0.020) | <0.001 | 0.735 (0.561 to 0.963) | 0.026 | 1.041 (0.744 to 1.458) | 0.813 |
| | | Diet based | −0.055 (−0.098 to −0.012) | 0.012 | 0.515 (0.339 to 0.785) | 0.002 | 0.504 (0.288 to 0.884) | 0.017 |
| | | Physical activity based | −0.018 (−0.068 to 0.032) | 0.472 | 0.660 (0.405 to 1.075) | 0.095 | 0.612 (0.314 to 1.193) | 0.149 |
| | | Control | Ref | | Ref | | Ref | |
| | Higher (n=3251) | Mixed approach | −0.053 (−0.069 to −0.037) | <0.001 | 0.947 (0.765 to 1.174) | 0.620 | 1.411 (1.056 to 1.884) | 0.020 |
| | | Diet based | −0.013 (−0.037 to 0.010) | 0.271 | 0.609 (0.437 to 0.849) | 0.003 | 0.962 (0.572 to 1.616) | 0.882 |
| | | Physical activity based | −0.018 (−0.040 to 0.004) | 0.111 | 1.030 (0.751 to 1.413) | 0.855 | 1.205 (0.761 to 1.909) | 0.427 |
| | | Control | Ref | | Ref | | Ref | |

Lower education: secondary education or less. Higher education: at least some tertiary education.
Mixed approach interventions included components of diet, physical activity, GWG monitoring and/or behaviour change.
*P value: Wald test.
†With fixed effects. Weight gain per week (kg/week) adjusted for body mass index (BMI) (kg/m²).
‡With fixed effects. Adjusted for maternal age, parity and ethnicity. Weight gain per week (kg/week) adjusted for BMI (kg/m²).

study is novel in that it is the first IPD meta-analysis, using a one-step approach for statistical analyses, in which data from individual pregnant women were used to estimate the effect of varying types of lifestyle interventions on GWG of women from different educational backgrounds.

### Clinical relevance of our findings

In practice, dietary interventions have the potential be used as a baseline intervention, offered to all pregnant women, to encourage appropriate weight gain. Women with secondary education or less could be offered mixed interventions (additional behaviour change counselling or physical activity advice), to complement the dietary intervention. Exercise alone interventions were not effective in managing weight gain in this analysis. The reduction in GWG associated with dietary and mixed interventions among women of low educational attainment can be considered clinically significant, given that if a woman was to follow the prescribed intervention for the duration of trimesters 2 and 3 (27 weeks) she would gain 1.49 and 1.19 kg less, respectively, than a woman following usual care.

The success of dietary interventions over other lifestyle interventions in reducing GWG may be attributed to many factors. The importance of a woman's diet in pregnancy has been documented for centuries[48]; thus, dietary changes are perhaps more acceptable than changes in physical activity behaviours. Pregnancy itself is an incentive for positive behaviour changes in dietary choices, but less so for physical activity.[16] Misconceptions regarding exercise during pregnancy remain, with many perceiving it to pose a safety risk for their baby,[49] and believing pregnancy is a time for rest.[50] In addition, women may experience feelings of tiredness, nausea and physical discomfort, as well as time commitments that prevent them from exercising.[49–51]

Mixed approach interventions increased the risk of inadequate weight gain among highly educated women. A potential hypothesis is that these women may already be aware of healthy lifestyle strategies, have a higher locus of control[52] and future salience,[53] and therefore may already monitor weight gain without intervention. Consequently, providing multiple approaches to reduce GWG could have a paradoxical effect. However, since 60.4% of highly educated women in this meta-analysis exceeded weight gain guidelines, the focus of healthcare professionals should be on meeting adequate GWG, rather than exerting excess caution in avoiding mixed approach interventions. It is reasonable to suggest that the dietary approach would be sufficient among those with tertiary education in reducing excessive GWG.

### Strengths and limitations

The pooling of individual patient data allows for the estimation of the differential effects of education and lifestyle interventions on weight gain across the included cohorts with greater power than individual RCTs.[19] IPD meta-analysis does not rely on aggregate data extracted from RCTs,

and allows for models to be adjusted for confounders.[21] Despite the congruence between our results and those in the wider literature, it must be highlighted that reporting of negative findings in lifestyle interventions is poor; thus, the evidence base to which we compare our findings may not reflect the true nature of lifestyle interventions in pregnancy. The data were however strengthened by the relatively low variability between studies. In addition, we analysed and reported findings using GWG/week, which enabled the comparison of studies that used different gestational ages for measuring GWG.

We acknowledge that a limitation of our work is the lack of prepregnancy weight measurements available, a methodological issue that affects much GWG research.[54] We classified BMI using self-reported prepregnancy weight, which may be subject to under-reporting and hence under-classification of BMI.[54] However, using the first trimester weight to determine prepregnancy BMI would also have been problematic due to weight gain in trimester 1 and potential for overclassification of BMI.[54] Although it has been suggested that prepregnancy weight can be somewhat accurately estimated using the first trimester weight, by 12 weeks' gestation weight gain can result in up to 9% of women being misclassified for BMI.[55] This was not a viable option since some of the first weight measurements were taken as late as 18 weeks. Early pregnancy weight was estimated using self-reported prepregnancy weight for a small number of participants (n=295, 5.7%).

While limitations of the IOM guidelines have been expressed,[56] they are the most widely used recommendations, and are comparable to Intergrowth 21st Project weight gain centiles for women with a healthy BMI[57] and new weight gain charts for women with a healthy BMI proposed by the Maternal Obesity and Childhood Outcomes (MOCO) collaboration.[58] We also used a novel method of extrapolating the IOM expected weight gain per week in trimester 1, and this enabled calculation of trimesters 2 and 3 wt gain per week. The maternal characteristics of women who were included in the IPD meta-analysis were different to those excluded; a finding that is likely related to the eligibility criteria of this meta-analysis which excluded all data from some studies that did not collect data pertaining to key characteristics required.

Lastly, it must be stated that although the results were statistically significant, the effect sizes were small; thus, the relevance of our findings in the clinical setting must be considered.

### CONCLUSIONS

In summary, pregnant women with lower educational attainment are at an increased risk of both excessive and inadequate weight gain highlighting the need for intervention among this vulnerable population of women. Given that dietary interventions do not widen health inequalities, are successful in promoting appropriate weight gain, and are acceptable, healthcare professionals

should consider implementing nutrition-based interventions as part of baseline maternity care packages for all pregnant women.

**Author affiliations**

[1]UCD Perinatal Research Centre, Obstetrics and Gynaecology, UCD School of Medicine, University College Dublin, Dublin, Ireland

[2]Centre for Support and Training in Analysis and Research (CSTAR), School of Public Health, Physiotherapy and Sports Science, University College Dublin, Dublin, Ireland

[3]Women's Health Research Unit, Barts and The London School of Medicine and Dentistry, Queen Mary University of London, London, UK

[4]Multidisciplinary Evidence Synthesis Hub (mEsh), Barts and The London School of Medicine and Dentistry, Queen Mary University of London, London, UK

[5]Department of Nutrition, Exercise and Sports, Univesity of Copenhagen, Copenhagen, Denmark

[6]Facultad de Ciencias de la Actividad Fisica y del Deporte (INEF), Universidad Politecnica de Madrid, Madrid, Spain

[7]Department of Development and Regeneration KU Leuven, University of Leuven, Leuven, Belgium

[8]Faculty of Health and Social Work, UC Leuven-Limburg, Leuven, Belgium

[9]Faculty of Medicine and Health Sciences, Centre for Research and Innovation in Care (CRIC), University of Antwerp, Belgium

[10]Obstetrics and Gynecology, School of Medical Sciences, University of Campinas, Campinas, Brazil

[11]School of Clinical and Experimental Medicine, College of Medical and Dental Sciences, University of Birmingham, Birmingham, UK

[12]Obstetrics and Gynaecology, Faculty of Medicine, VU University Medical Center, Amsterdam, The Netherlands

[13]Department of Obstetrics and Gynecology, Universitaire Ziekenhuizen Leuven, Leuven, Belgium

[14]Obstetrics and Gynaecology, School of Paediatrics and Reproductive Health, The Unversity of Adelaide, Adelaide, Australia

[15]Department of Obstetrics and Gynecology, Alexandria University, Alexandria, Egypt

[16]Mother-Infant Department, University of Modena and Reggio Emilia, Modena, Italy

[17]Clinical Nutrition Research, Copenhagen University Hospital Herlev-Gentofte, Gentofte, Denmark

[18]School of Human Sciences, The University of Western Australia, Perth, Australia

[19]Department of Sports Medicine, Norwegian School of Sport Sciences, Oslo, Norway

[20]Monash Centre for Health Research and Implementation, School of Public Health and Preventive Medicine, Monash University, Melbourne, Victoria, Australia

[21]Else Kroener-Fresenius-Center for Nutritional Medicine, Klinikum rechts der Isar, Technische Universität München, Munich, Germany

[22]Department of Endocrinology, Odense University Hospital, University of Southern Denmark, Odense, Denmark

[23]Health Sciences, Faculty of Social Sciences, University of Tampere, Tampere, Finland

[24]Robinson Institute, School of Paediatrics and Reproductive Health, The University of Adelaide, Adelaide, Australia

[25]Department of Public Health and General Practice, Faculty of Medicine, Norwegian University of Science and Technology, Trondheim, Norway

[26]Clinical Services, St. Olavs Hospital, Trondheim University Hospital, Trondheim, Norway

[27]Midwifery Department, Faculty of Nursing and Midwifery, Mazandaran University of Medical Science, Sari, Iran

[28]Kinesiology Department, College of Science and Mathematics, California Polytechnic State University, San Luis Obispo, California, USA

[29]Department of Women and Children's Health, School of Life Course Sciences, King's College London, London, UK

[30]Nutrition Information and Knowledge Transfer, Competence Centre for Nutrition (KErn), Freising, Germany

[31]Department of Obstetrics and Gynecology, Hvidovre Hospital, University of Copenhagen, Copenhagen, Denmark

[32]Obstetric Clinic, Juliane Marie Centret, Rigshospitalet, University of Copenhagen, Copenhagen, Denmark

[33]Department of Obstetrics and Gynecology, Academisch Medisch Centrum Universiteit van Amsterdam, Amsterdam, The Netherlands

[34]Department of Obstetrics and Gynecology, Sorlandet Hospital, Kristiansand, Norway

[35]Department of Obstetrics and Gynaecology, St. Olavs Hospital, Trondheim University Hospital, Trondheim, Norway

[36]Department of Laboratory Medicine Children's and Women's Health, Norwegian University of Science and Technology, Trondheim, Norway

[37]Department of Management and Health Care, Universidade Federal de Sao Paulo, São Paulo, Brazil

[38]Department of Internal Medicine, University of Manitoba College of Medicine, Winnipeg, Canada

[39]Obstetrics and Gynaecology, University of Melbourne, Victoria, Australia

[40]Department of Obstetrics and Gynecology, Oslo University Hospital, Oslo, Norway

[41]Department of Public and Occupational Health, VU University Medical Center, Amsterdam, The Netherlands

[42]Department of Obstetrics and Gynecology, Odense University Hospital, University of Southern Denmark, Odense, Denmark

[43]School of Nursing, University of North Carolina at Chapel Hill, Chapel Hill, North Carolina, USA

**Acknowledgements** The authors are also grateful to the i-WIP Network for the individual participant data shared and to Helena Teede, Janette Khoury and Márcia Vitolo for the contribution of data.

**Contributors** ECOB, FMMcA: wrote a proposal to the i-WIP data access committee; set the objectives, study design and data analysis plan. ECOB, FMMcA, RS, AAG and GA: analysis and interpretation of data; drafting of manuscript; critical revision; final approval of version to be published; agreeable to be accountable for all aspects of the work. ER: i-WIP IPD database manager; study conception and design; critical revision; final approval of version to be published; agreeable to be accountable for all aspects of the work. AA, RBC, AB, JGC, AC, CJMdG, RD, JMD, NEB, FF, NG, KG, LH, CH, HH, DMJ, KK, TIK, RL, BWM, SM, NM-T, JAO, MP, EP, SP, LP, KR, GR, KMR, AER, LS, KÅS, TTS, GS, AS, SNS, FGS, ShT, SeT, MNMvP, CV, IV and SY: study conception and design; critical revision; final approval of version to be published; agreeable to be accountable for all aspects of the work.

**Funding** The data come from a National Institute for Health Research (NIHR) funded project (HTA-12/01/50) and the Queen Mary University of London is its legal sponsor. The database was initially funded by the NIHR and is currently funded by WHO. This work was also supported by the Health Research Board, Health Research Centre for Health and Diet Research, Ireland.

**Competing interests** None declared.

**Patient consent for publication** Not required.

**Ethics approval** IPD meta-analysis. Research ethics was approved for each study included.

**Provenance and peer review** Not commissioned; externally peer reviewed.

**Data sharing statement** The Queen Mary University of London is the legal custodian of the i-WIP dataset. The access to the dataset is regulated by prespecified terms and conditions (available on request) and overseen by Data Access Committee. For further information visit: https://iwipgroup.wixsite.com/collaboration/related-projects

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
