## [Reviewer comments · BMJ Open]

ARTICLE DETAILS

TITLE (PROVISIONAL)	Impact of maternal education on response to lifestyle interventions to reduce gestational weight gain - Individual participant data meta-analysis
AUTHORS	O'Brien, Eileen; Segurado, Ricardo; Geraghty, Aisling; Alberdi, Goiuri; Rogozinska, Ewelina; Astrup, Arne; Barakat Carballo, Rubenomar; Bogaerts, Annick; Cecatti, Jose; Coomarasamy, Arri; de Groot, Christianne; Devlieger, R.; Dodd, Jodie; El Beltagy, Nermeen; Facchinetti, Fabio; Geiker, Nina; Guelfi, Kym; Haakstad, Lene; Harrison, Cheryce; Hauner, Hans; Jensen, DM; Khan, Khalid; Kinnunen, Tarja; Luoto, Riitta; Willem Mol, Ben; Mørkved, Siv; Motahari-Tabari, Narges; Owens, Julie; Perales, Maria; Petrella, Elisabetta; Phelan, Suzanne; Poston, Lucilla; Rauh, Kathrin; Rayanagoudar, Girish; Renault, KM; Ruifrok, A; Sagedal, Linda; Salvesen, Kjell; Scudeller, Tania; Shen, Gary; Shub, Alexis; Stafne, Signe; Surita, Fernanda; Thangaratinam, Shakila; Tonstad, Serena; van Poppel, Mireille; Vinter, Christina; Vistad, Ingvild; Yeo, SeonAe; McAuliffe, Fionnuala

VERSION 1 - REVIEW

REVIEWER	Dr Lesley MacDonald-Wicks the University of Newcastle, Australia
REVIEW RETURNED	17-Oct-2018

GENERAL COMMENTS	Thank you for the opportunity to review this paper. There is a need for a personalised approach to GWG in women, particularly based on aspects such as SES or educational level. The paper is well presented. I have a couple of minor issues that need to be clarified: 1. The title states education level but the aim states the focus is on SES. There is a statement of maternal education level being the marker of SES, but I think this should still be consistent. Please consider changing the title to reflect the aim accurately.2. There is no statement of ethical clearance for the included studies. This is likely in the main SR paper but should be stated in this paper as well (eg: all papers included in the review had ethical clearances from relevant HRE committees)3. There are small grammatical errors throughout the paper, please review the paper for these.4. The lifestyle interventions are not the focus of this paper, however the conclusion is that diet interventions reduce GWG. The reporting of lifestyle interventions is notoriously poor in the literature. If this SR would like to suggest the inclusion of diet interventions to manage both adequate and inadequate weight
--

	gain in pregnancy then acknowledging the limitations of reported diet interventions should be included.
--	---

REVIEWER	Defu Ma School of Public Health, Peking University Health Science Center
REVIEW RETURNED	17-Oct-2018

GENERAL COMMENTS	This paper is an individual participant data meta-analysis to identify if socioeconomic status is a prognostic factor for gestational weight gain (GWG), and to determine the differential effects of lifestyle interventions (diet-alone, exercise-alone or mixed approach) on gestational weight gain, according to socioeconomic status. And the results are very interesting. According to the paper, pregnant women with lower education are at an increased risk of excessive and inadequate GWG, and diet-alone interventions seem the most appropriate choice for these women Although the risk factors about gestational weight gain have been investigated by a lot of studies, this paper is very useful because of its powerful proof. This paper has higher application value for pregnancy education. There is one problem in this paper. Some tables are not three-line tables. The author should revise it.
--

REVIEWER	David Rehkopf Stanford University, United States of America
REVIEW RETURNED	08-Jan-2019

GENERAL COMMENTS	The data used and the analysis approach are excellent, and the paper uses these to draw appropriate conclusions about the impact of education on heterogenous treatment effects. I have only minor suggestions. -A large number of individual participants are dropped due to missing data. While the inclusion of the RCTs is not representative of any particular target population, it would still be useful to know whether those individuals excluded differed in terms of key demographic and study outcome characteristics. This could be presented as a supplementary table. -The study seems to suggest two types of questions, whether low income women benefit from the interventions, and whether there are differences by level of education. These are distinct study questions, and both are important. I would suggest that when describing the findings in both the abstract and the results section, these two questions are specified more clearly.
---

REVIEWER	Josie Athens University of Otago, New Zealand.
REVIEW RETURNED	28-Jan-2019

GENERAL COMMENTS	Authors declare, both in the abstract and introduction, that their objective is to look at the effect of socioeconomic status on gestational weight gain (GWG). However, they use educational attainment as a proxy. I like that all statements in the rest of the paper, particularly about results and conclusions, are about education and not socioeconomic status. I wish that would be clear in the abstract, as to not mislead readers. The fact that they use an individual participant data (IPD) meta-analysis is not an excuse to not present a forest plot and publication bias results from the papers. They include, as one of the supplementary tables, means and standard deviations for each study stratified by educational attainment; I would like to see the actual differences in GWG. For GMG as an outcome, authors don't report the value of the coefficient of determination, how good are their models? Is variability in the data explained by them? By how much? My main concern is about the small effects obtained from the models. Clinical differences should be selected apriori. They discuss about those small effects and however conclude that indeed the effect was significant. It is clear that their significant p-values are the result of their sample sizes, it has to be clear what the literature or they consider as a clinically significant difference.
---

VERSION 1 – AUTHOR RESPONSE

Reviewer: 1

Reviewer Name: Dr Lesley MacDonald-Wicks

Institution and Country: the University of Newcastle, Australia

Comment	Revision
Thank you for the opportunity to review this paper. There is a need for a personalised approach to GWG in women, particularly based on aspects such as SES or educational level. The paper is well presented.	Many thanks for your comment.
1. The title states education level but the aim states the focus is on SES. There is a statement of maternal education level being the marker of SES, but I think this should still be consistent. Please consider changing the title to reflect the aim accurately.	Thank you for this suggestion. We have changed the phrase socioeconomic status to educational attainment throughout the manuscript.
2. There is no statement of ethical clearance for the included studies. This is likely in the main SR paper but should be stated in this paper as well (eg: all papers included in the review had ethical clearances from relevant HRE committees)	Ethical approval was not sought for the IPD meta-analysis. The following text has been added to the methods section: All individual trials included in this IPD meta-analysis had ethical approval from relevant ethical committees.
3. There are small grammatical errors throughout the paper, please review the paper for these.	The manuscript has been reviewed and checked for grammatical and spelling errors.

4. The lifestyle interventions are not the focus of this paper, however the conclusion is that diet interventions reduce GWG. The reporting of lifestyle interventions is notoriously poor in the literature. If this SR would like to suggest the inclusion of diet interventions to manage both adequate and inadequate weight gain in pregnancy then acknowledging the limitations of reported diet interventions should be included.	Thank you for this comment, we have added the following statement to the discussion section. Despite the congruence between our results and those in the wider literature, it must be highlighted that reporting of negative findings in lifestyle interventions is poor, thus the evidence base to which we compare our findings may not reflect the true nature of lifestyle interventions in pregnancy.
---	--

Reviewer: 2

Reviewer Name: Defu Ma

Institution and Country: School of Public Health, Peking University Health Science Center

Comment	Revision
There is one problem in this paper. Some tables are not three-line tables. The author should revise it.	All tables in the main text document have been updated to include only a line above the headings, below the headings, and above/below the footnote

Reviewer: 3

Reviewer Name: David Rehkopf

Institution and Country: Stanford University, United States of America

Comment	Revision
A large number of individual participants are dropped due to missing data. While the inclusion of the RCTs is not representative of any particular target population, it would still be useful to know whether those individuals excluded differed in terms of key demographic and study outcome characteristics. This could be presented as a supplementary table.	Many thanks for this comment. Table S4 has been added to the supplementary files that compares the characteristics of women included in the IPD and those excluded from the IPD.
The study seems to suggest two types of questions, whether low income women benefit from the interventions, and whether there are differences by level of education. These are distinct	Many thanks for your comment. We realise now that our second question was not phrased clearly. We aimed to identify which intervention would be most effective among women of low education and among women of high education. We felt that as clinicians, if for example we were advising a woman of low socioeconomic status to monitor her gestational weight gain, we

study questions, and both are important. I would suggest that when describing the findings in both the abstract and the results section, these two questions are specified more clearly.	would want to know which intervention would suit her most. We did not aim to compare the effectiveness of different types of intervention across educational levels. To address this, we have changed the phrasing of our second research aim from “according to education attainment” to “stratified by educational attainment”.
---	--

Reviewer: 4

Reviewer Name: Josie Athens

Institution and Country: University of Otago, New Zealand.

Comment	Revision
Authors declare, both in the abstract and introduction, that their objective is to look at the effect of socioeconomic status on gestational weight gain (GWG). However, they use educational attainment as a proxy. I like that all statements in the rest of the paper, particularly about results and conclusions, are about education and not socioeconomic status. I wish that would be clear in the abstract, as to not mislead readers.	Thank you for this suggestion. As per our response to reviewer one, we have changed the phrase socioeconomic status to educational attainment throughout the manuscript.
The fact that they use an individual participant data (IPD) meta-analysis is not an excuse to not present a forest plot and publication bias results from the papers. They include, as one of the supplementary tables, means and standard deviations for each study stratified by educational attainment; I would like to see the actual differences in GWG	Thank you for this suggestion. In terms of publication bias, a funnel plot has been created to assess the small study effects on trials in the IPD meta-analysis of educational attainment and gestational weight gain per week. See Figure S1 in the supplementary files. The even distribution of effect estimates and standard error indicate low bias in publications. A Forest plot has been created and submitted with the revision as Figure S2. Mean differences and confidence intervals for the difference in gestational weight gain between the intervention and control, stratified by study and educational attainment, have been added to Table S3.
For GWG as an outcome, authors don't report the value of the coefficient of determination, how good are their models? Is variability in the data explained by them? By how much?	This analysis method uses a general linear mixed model with random effects, thus a coefficient of determination (R^2) is not calculated as standard in a general linear model. Instead, to assess the magnitude of study variability in the outcomes, we calculated an intra-class correlation coefficient (ICC) from the fitted models, interpretable as an index of heterogeneity such as the I^2. The intra-class correlation coefficient for the study

	(ICC_{study}) relates to the proportion of variability due to cohort-to-cohort variation, and the intra-class correlation coefficient for the intervention (ICC_{intervention}) relates to the proportion of variability in the intervention treatment effect. The literature does not recommend R² above ICC for general linear mixed models, and methods for estimating R² are still evolving (Nakagawa et al., 2017) In our results section, we describe the ICC results. The unadjusted linear mixed-models found that women with lower educational attainment gained significantly more weight per week (kg) compared to those with tertiary education (B = 0.014), however this association was not significant when the model was adjusted for maternal age, parity and ethnicity (Table 4a). The ICC_{study} was 0.1% and ICC_{intervention} was 0.0%. Women of lower educational attainment had an increased risk of both excessive (OR: 1.182) and inadequate (OR: 1.284) GWG as per the IOM guidelines, controlling for intervention, maternal age, parity, ethnicity, with random effects (Table 5b). The ICC calculations for model 2 of Table 4 (b) were low; excessive weight gain: ICC_{study} 8.1% and ICC_{intervention} 0.4%; inadequate weight gain: ICC_{study} 5.0% and ICC_{intervention} 0.7%. Reference: Nakagawa S, Johnson PCD, Schielzeth H. The coefficient of determination R² and intra-class correlation coefficient from generalized linear mixed-effects models revisited and expanded. J R Soc Interface [Internet]. The Royal Society; 2017;14(134):20170213.
My main concern is about the small effects obtained from the models. Clinical differences should be selected apriori. They discuss about those small effects and however conclude that indeed the effect was significant. It is clear that their significant p-values are the result of their sample sizes, it has to be clear what the literature or they consider as a clinically significant difference.	Despite the lead authors being clinicians working in clinical practice with pregnant women, we found it difficult to decide what level of weight gain is considered a clinically significant. We know from the literature that gaining weight in excess of the IOM guidelines increases the risk of a large for gestational age infant, gestational diabetes, caesarean section, and maternal weight retention. One finding from our data suggests that dietary interventions reduce GWG/week by 0.055kg among women with secondary education or less, equating to 1.49 kg less over the 27 weeks in trimester 2 and 3. This reduction in weight gain could be quite significant in maintaining some women in the normal GWG range, given that weight gain/week recommendation is 0.2kg/week for women with BMI>30 (total weight gain 5 – 9kg). We have added the following text to the discussion (clinical relevance of our findings): The reduction in gestational weight gain associated with dietary and mixed interventions among women of low educational attainment can be considered clinically significant, given that if a woman was to follow the prescribed intervention for the duration of trimester two and three (27 weeks) she would gain 1.49 and 1.19 kg less, respectively, than a woman following usual care.

VERSION 2 – REVIEW

REVIEWER	Dr Lesley MacDonald-Wicks The University of Newcastle, Australia
REVIEW RETURNED	01-Apr-2019

GENERAL COMMENTS	Dear Authors Thank you for the opportunity to review this important piece of research. Weight gain in pregnancy is of vital importance. This paper provides insight into the utility of diet and/or PA interventions on GWG. I have a small number of questions that I think need to be answered about the methodology of the IPD.  1. While I acknowledge that the i-WIP collaboration paper has been published elsewhere I think a brief statement of the way the collaboration was established and the number of trials approached and agreed to be included should be made. 2. Detail of the success of randomisation of each of the trials is needed. Is any data from non-randomised or inadequately randomised patients included? 3. Was analysis undertaken by trial, was there any trial effects, especially effects of small sample sizes. 4. A funnel plot was presented that indicated little publication bias, however were there any non published data included? I clinical relevance section, diet is recommended for all pregnant women, but for those of low educational attainment additional PA is suggested. There is no evidence to support PA as a strategy for GWG, please justify it's inclusion in the recommendations. small grammatical consideration in line 12/13 in introduction, sentence starting with Currently... the words 'remain to be important determinants' is included. This does not make sense.
--

REVIEWER	Josie Athens University of Otago
REVIEW RETURNED	21-Mar-2019

GENERAL COMMENTS	I am pleased by the way authors address concerns from all reviewers in a clear and succinct way.
--

VERSION 2 – AUTHOR RESPONSE

Reviewer: 1

While I acknowledge that the i-WIP collaboration paper has been published elsewhere I think a brief statement of the way the collaboration was established and the number of trials approached and agreed to be included should be made.	Many thanks for this comment. The following sentence had been edited and expanded in the introduction: The International Weight Management in Pregnancy (i-WIP) Individual Patient Data Collaborative Network was established in 2013 to determine the differential effects of weight
--	---

	management interventions in pregnancy on maternal weight gain and pregnancy outcomes (20). The collaboration between researchers from 16 countries has resulted in a repository of data from 36 randomised trials with over 60 variables and individual data from more than 12,500 pregnant women.
2. Detail of the success of randomisation of each of the trials is needed. Is any data from non-randomised or inadequately randomised patients included?	All data included in the i-WIP network are from randomised controlled trials. No non-randomised or inadequately randomised patients were included. The following has been added to the IPD integrity and risk of bias assessment paragraph within the Methods section. The randomisation ratio, baseline characteristics and method of analysis provided by the original study teams to the i-WIP Network were compared with the published information. Any discrepancies were queried and rectified as necessary with input from the original authors.
3. Was analysis undertaken by trial, was there any trial effects, especially effects of small sample sizes.	We thanks the reviewer for this comment. Analysis was carried out by individual trial in “Table S3. Individual Study Results – Gestational Weight Gain per Week”. This table is referred to in the results section on page 13. A funnel plot was created to assess the small study effects on trials in the IPD meta-analysis of educational attainment and gestational weight gain per week. (Figure S1). Statistical analysis to determine the effect of small studies was considered in the main i-WIP publication, but was not repeated for this subset analysis. The following text is available in the main i-WIP publication: “We found visual and statistical evidence (Egger’s test $P=0.04$) of small study effects in the contour enhanced funnel plots for the IPD meta-analysis of the overall effect on gestational weight gain. The asymmetry of the plot was not improved by the addition of study level data from non-IPD studies to the meta-analysis. When studies with high risk of bias were excluded from the analysis, the symmetry of the funnel plot improved (Egger’s test $P=0.61$). We found significant evidence of small study effects for the maternal composite outcome (Peter’s test $P=0.04$), but not for the offspring composite outcome ($P=0.85$) (see web appendix 4).” The International Weight Management in Pregnancy (i-WIP) Collaborative Group. Effect of diet and physical activity based interventions in pregnancy on gestational weight gain and

	pregnancy outcomes: meta-analysis of individual participant data from randomised trials. BMJ. 2017;358:j3119
4.A funnel plot was presented that indicated little publication bias, however were there any non published data included?	Many thanks for this comment. No non-published data were included in this sub-set analysis.
In the clinical relevance section, diet is recommended for all pregnant women, but for those of low educational attainment additional PA is suggested. There is no evidence to support PA as a strategy for GWG, please justify it's inclusion in the recommendations.	We thank the reviewers for this comment. Both the dietary interventions and mixed approach interventions (diet with behaviour change and/or physical activity) significantly improved weight gain among women with low educational attainment. Physical activity alone did not seem to affect weight gain. We have changed the text to improve the clarity of our message to: Women with secondary education or less could be offered mixed interventions (additional behaviour change counselling or physical activity advice), to complement the dietary intervention. Physical activity alone interventions were not effective in managing weight gain in this analysis.
small grammatical consideration in line 12/13 in introduction, sentence starting with Currently... the words 'remain to be important determinants' is included. This does not make sense.	Many thanks for highlighting this change. We have amended the sentence to the following: The current situation is that maternal social, economic and educational backgrounds are key determinants of health and lifestyle behaviours among pregnant women across the world